# AAV2-driven endothelin induces chronic reduced retinal blood flow/retinal ganglion cell loss in rats

Ge Shi[1,*], Kota Sato[1,2,*] , Nana Takahashi[1], Michiko Ohno-Ohishi[1], Namie Murayama[1], Chiaki Yamaguchi[1],
Daisuke Saigusa[3,4], Toru Nakazawa[1,2,5,6]

**Dysfunction of ocular blood flow (BF) is believed to be one of the causes of glaucomatous pathology. However, whether this dysfunction is indeed a cause or is actually a consequence of optic nerve degeneration remains controversial. Here, we established a new animal model of chronic BF reduction in the retina to mimic glaucoma. We found that retinal BF in rats, as measured with laser speckle flowgraphy, was significantly reduced 3 wk after an intravitreal injection of AAV2-human endothelin-1 (AAV2-hEDN1). The number of retinal ganglion cells was also reduced in rats that received AAV2-hEDN1 injection. Immunostaining signals for GFAP and the endothelin-B receptor were enhanced in the rat retinas after AAV2-hEDN1 injection. Moreover, mRNA levels of _Ripk1_/ _Ripk3_ and _Tnf_ in the retina increased, and glutathione levels in the aqueous humor decreased in rats that received AAV2-hEDN1 injection. Our findings demonstrate that endothelin-induced chronic retinal BF reduction leads to increased astrocyte activation and oxidative stress, which in turn induces retinal ganglion cell necroptosis. This suggests that methods to improve ocular BF have potential as novel therapies for glaucoma.**

## Introduction

Glaucoma, the second leading cause of blindness worldwide, is characterized by progressive optic neuropathy that can lead to irreversible vision loss (Quigley, 2005). Glaucoma is primarily associated with the loss of retinal ganglion cells (RGCs) and their axons (Quigley et al, 1995). Ocular blood flow (BF) is considered to be a cause of these neurodegenerative changes (Shen et al, 2023). Ocular BF is an essential factor in the development and progression of glaucoma, involving not only local ocular BF but also systemic BF (Flammer et al, 2002). Interestingly, patients diagnosed with glaucoma have consistently exhibited reduced BF in major arteries, such as the ophthalmic artery, central retinal artery, and posterior ciliary arteries, even at the time of diagnosis (Broadway & Drance, 1998; Flammer & Orgül, 1998; Flammer et al, 2002; Emre et al, 2004). Laser speckle flowgraphy (LSFG) is a non-invasive and highly reproducible technique (Sugiyama et al, 2010). A meta-analysis and our previous report demonstrated that LSFG-measured retinal BF decreased in patients with open-angle glaucoma (OAG) (Kiyota et al, 2018, 2021; Gu et al, 2021; Yamaguchi et al, 2024). Interestingly, in human patients with OAG, a reduction in retinal BF occurs from the earliest stages of the disease (Shiga et al, 2016).

Endothelin-1 (ET-1), one of three endothelin isoforms, is the most potent vasoconstrictor in the human body (Yanagisawa et al, 1988; Inoue et al, 1989) and the most potent contributor to the pathophysiology and dysfunction of retinal BF in glaucoma. It is known that intravitreal injection of ET-1 in rodent models causes RGC degeneration and that ET-1 levels are a factor in glaucoma development (Bursell et al, 1995; Chauhan et al, 2004; Masuzawa et al, 2006; Lau et al, 2006; Wang et al, 2008; Nagata et al, 2014; Lommatzsch et al, 2022). However, ET-1 intravitreal injection, as an example, has the drawback as a model that it does not reflect glaucomatous vascular dysfunction. Indeed, Taniguchi et al showed that although ET-1 injection promoted acute and severe vascular contraction, this had recovered to baseline 7 d after the ET-1 intravitreal injection (Taniguchi et al, 2006). Therefore, it remains unclear to what extent ocular perfusion defects in animal models in previous studies are similar to those of actual patients. Because glaucoma, including OAG, generally has a chronic course, it is necessary to establish a chronic, moderate model that mimics the reduction in ocular BF in animal models, as well as in humans, to further clarify the pathophysiology of glaucoma.

Recently, Nordahl et al reported that AAV2-derived ET-1 can cause continuous mild vascular dysfunction, resulting in retinal degeneration (Nordahl et al, 2023). However, the use of LSFG in animal models of ocular BF disorders and the relationship with actual RGC loss have not been fully investigated. Thus, in this study,

[1]Department of Ophthalmology, Tohoku University Graduate School of Medicine, Miyagi, Japan   [2]Department of Advanced Ophthalmic Medicine, Tohoku University Graduate School of Medicine, Miyagi, Japan   [3]Laboratory of Biomedical and Analytical Sciences, Faculty of Pharmaceutial Science, Teikyo University, Tokyo, Japan [4]Department of Integrative Genomics, Tohoku Medical Megabank Organization, Tohoku University, Miyagi, Japan   [5]Department of Retinal Disease Control, Tohoku University Graduate School of Medicine, Miyagi, Japan   [6]Department of Ophthalmic Imaging and Information Analytics, Tohoku University Graduate School of Medicine, Miyagi, Japan

Correspondence: toru.nakazawa.e1@tohoku.ac.jp
*Ge Shi and Kota Sato contributed equally to this work

we attempted to establish a new animal model of dysfunctional retinal BF to mimic the condition of patients with glaucoma who have a moderate, chronic reduction in retinal BF. Moreover, we evaluated whether this chronic reduction in retinal BF involved RGC loss and a change in glial response, which is suggested to contribute to endothelin stimulation in the retina (Prasanna et al, 2011).

# Results

### Evaluation of different AAV capsids and promoters for the retinal surface layer

To determine the optimal AAV capsids and promoters to drive the target gene, we tested the transduction efficiency in the retina of different vectors and promoters. At first, we injected an AAV2-CMV-EGFP vector, AAV2-CAG-EGFP vector, AAV2-QuadYF-CMV-EGFP vector, and AAV2-QuadYF-CAG-EGFP vector into the vitreous of the rats. AAV2-CAG-EGFP led to the highest expression of EGFP in the retinas, especially in the GCL (Fig S1A). The signal from EGFP-driven AAV2-CAG was co-localized with RNA binding protein with multiple splicing (RBPMS) immunostaining (Fig S1B), suggesting that AAV2-CAG-EGFP infected the RGCs and drove the target gene.

### AAV2-driven endothelin-1 induced mild vasoconstriction in rat retinas

The canonical role of the endothelin system is to mediate vasoconstriction (Marola et al, 2020). First, we measured retinal BF using LSFG to evaluate the effect of intravitreal injection of ET-1. Compared with vehicle injection, retinal BF measured with MBR was reduced by ~13% for 20 min after ET-1 injection (Fig 1A and B). Next, to determine whether AAV2-CAG-hEDN1 led to retinal vasoconstriction, we used LSFG to measure whether there was a sustained expression of human *EDN1*. We observed no significant differences in BF 2 wk (92% compared with control) after the intravitreal injection of AAV2-CAG-hEDN1 or a control vector. In AAV2-CAG-hEDN1 group, however, MV was significantly reduced, by 74.3% 3 wk after intravitreal injection and 72.4% 4 wk after intravitreal injection, compared with the control vector (Fig 2A and B). To further confirm that human *EDN1* was induced, we measured transcriptional levels with a real-time PCR analysis. In the 4-wk AAV2-CAG-hEDN1 intravitreal injection group, the level of human *EDN1* was clearly elevated, by 278-fold (Fig 2C). These findings suggest that intravitreal injection of ET-1 immediately and dramatically reduced retinal BF, while AAV2-derived exogenous endothelin-1 chronically and gently reduced retinal BF, as in glaucoma patients.

### RGC loss in AAV2-CMV-hEDN1 rat retinas

Previous research has demonstrated that endothelin exposure causes RGC death (Blanco et al, 2017; Marola et al, 2020). To evaluate whether sustained endothelin stimulation and moderate dysfunction of retinal BF also modulate RGC loss, we examined RGCs with RBPMS immunostaining in AAV2-CAG-hEDN1-injected retinas. In the group of retinas observed 2 wk after AAV2-CAG-hEDN1

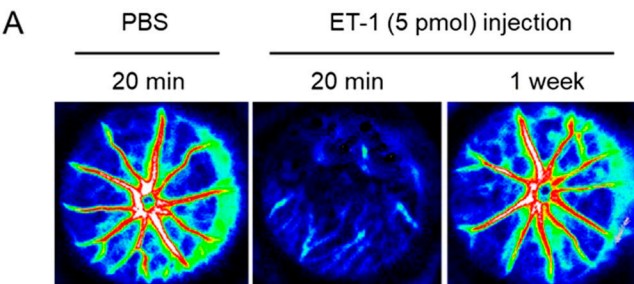
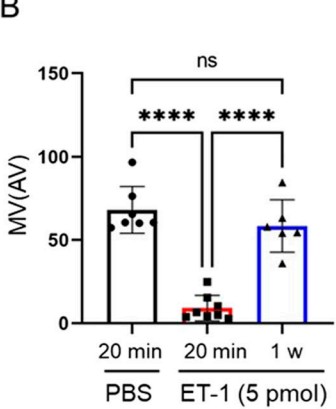

**Figure 1. Temporal blood flow reduction in rat retinas following ET-1 intravitreal injection.**
**(A)** Representative LSFG color maps showing, from left to right, PBS injected as a negative control, 20 min after intravitreal injection of ET-1, and 1 wk after intravitreal injection of ET-1. **(B)** Histogram showing the quantitative measurement of %MV 20 min after the intravitreal injection of PBS (n = 7), 20 min after the intravitreal injection of 5 pmol ET-1 (n = 8), or 1 wk after the intravitreal injection of 5 pmol ET-1 (n = 6). Data are expressed as mean ± SD (n = 3) (**$P <$ 0.01, ***$P <$ 0.001, ****$P <$ 0.0001 in a one-way ANOVA followed by the Tukey–Kramer test).

injection, there was no significant change in the distribution or quantity of RGCs compared with the group of retinas injected with AAV2-CAG-stuffer as a control vector (Fig S2A–D). On the other hand, in the group of retinas 4 wk after intravitreal injection of AAV2-CAG-hEDN1, the density of RGCs was significantly reduced, in the peripheral retinal area (79.4%, $P = 0.008$) compared with control group. The density of RGCs was also reduced in the near-optic-disc area (92.6%, $P = 0.036$) and central retinal area (90.4%, $P = 0.321$) in the AAV2-CAG-hEDN1 injection group 4 wk after intravitreal injection (Fig 3A–D). These findings suggest that chronic and moderate reduction in retinal BF causes RGC loss in rats, especially in the peripheral area of the retina.

### Astrocyte activation in AAV2-CAG-hEDN1 rat retinas

The astrocytes are the main glial cells of the central nervous system, including in the optic nerves (Tonari et al, 2012). When the endothelin expression level increases, proliferation of astrocytes and activation was found in retina (Gadea et al, 2008; Murphy et al, 2011; Prasanna et al, 2011; Alrashdi et al, 2018). Thus, we evaluated astrocyte activation with glial fibrillary acidic protein (GFAP)

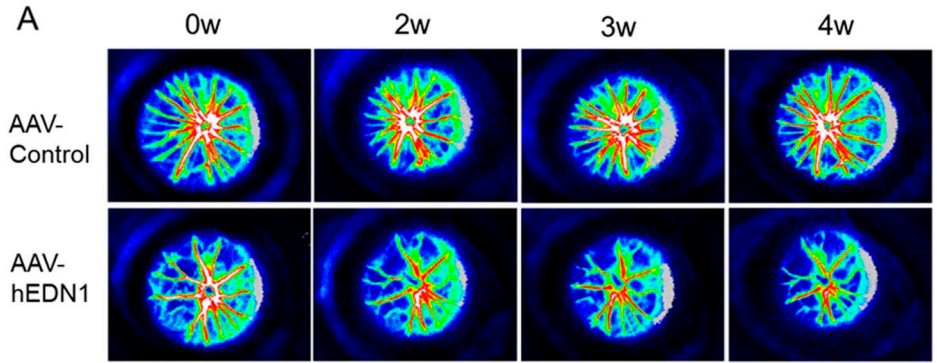

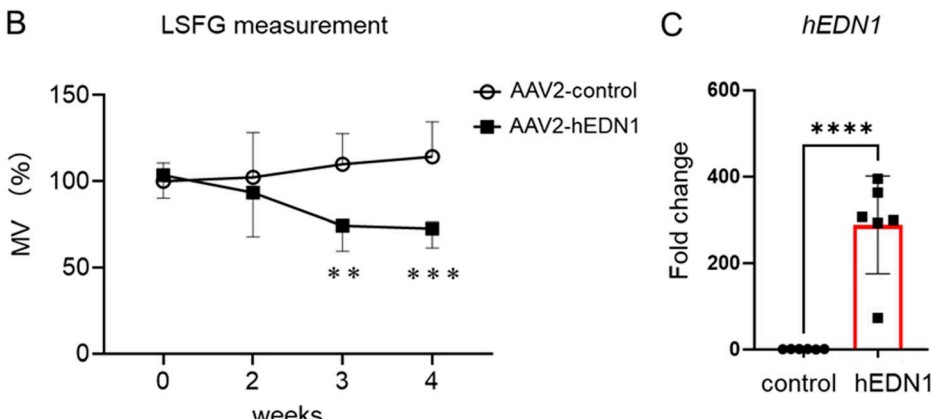

**Figure 2. AAV2-driven endothelin-1 expression modulates sustained blood flow reduction in rat retinas.**
**(A)** Representative LSFG color maps after the intravitreal injection of AAV2-CAG-hEDN1 or a control vector. **(B)** Quantitative graph of MV measured over time after injection of AAV-CAG-hEDN1 and control vectors into the retina. The data are expressed as mean ± SD (n = 6–15). **(C)** Transcriptional levels of human endothelin-1 (h*EDN1*) in rat retinas 4 wk after AAV-CAG-hEDN1 or control vector injection. Data are expressed as mean ± SD (n = 6). *$P < 0.05$, **$P < 0.01$, ***$P < 0.001$, ****$P < 0.0001$.

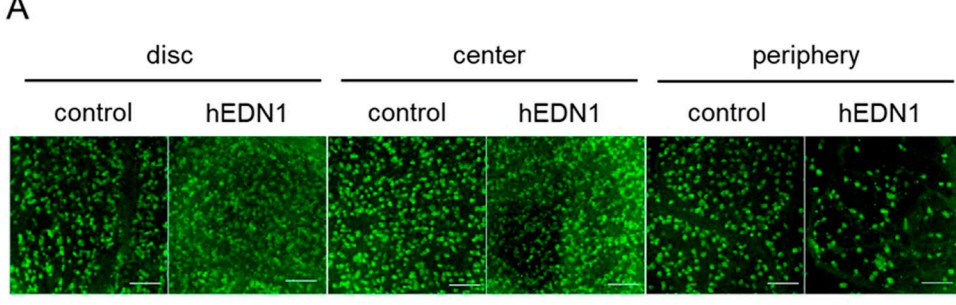

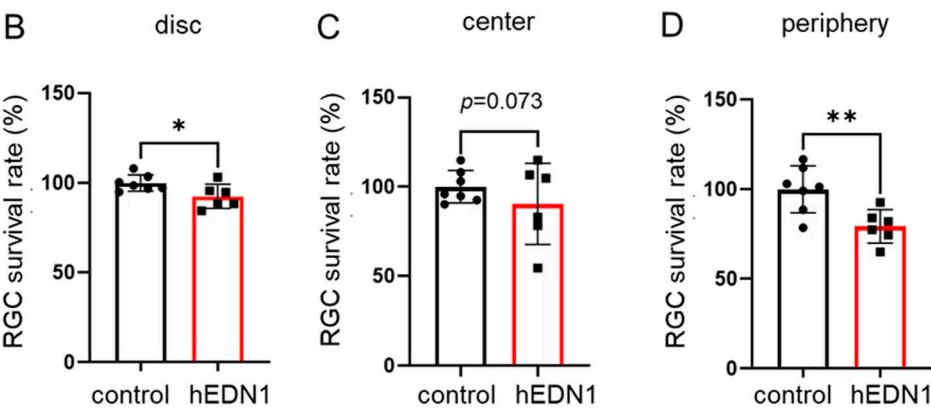

**Figure 3. RGC loss in AAV2-CMV-hEDN1 rat retinas 4 wk after intravitreal injection.**
**(A)** Representative RBPMS immunostaining images 4 wk after intravitreal injection of AAV2-CMV-hEDN1 or a control vector in the labeled retinal areas. **(B, C, D)** Histogram showing the ratio of RGC density in near disc (B), center (C), and peripheral (D) retinal area. Data are expressed as mean ± SD (n = 6–7). *$P < 0.05$. **$P < 0.01$. Scale bar: 100 μm.

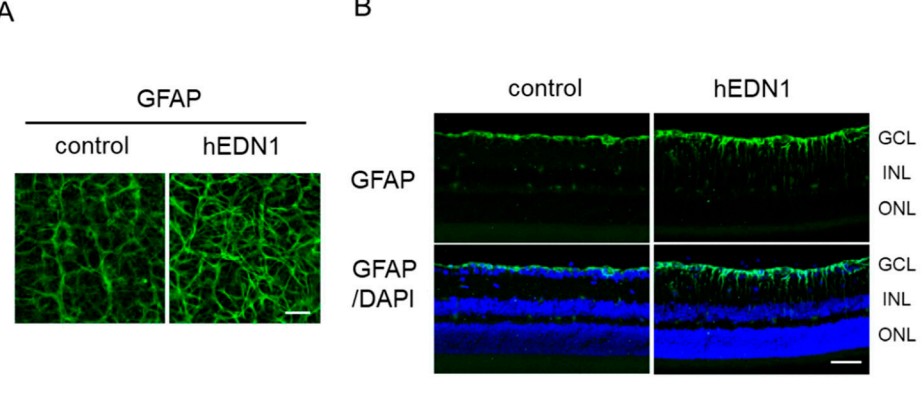

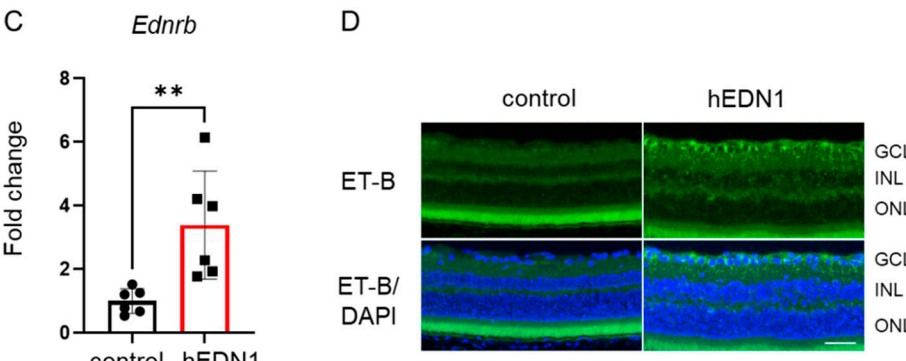

**Figure 4. Astrocyte activation in AAV2-CAG-hEDN1 rat retinas.**
**(A)** Retinal flat mount with immunostaining of GFAP in AAV2-CAG-hEDN1 rat retinas 4 wk after the intravitreal injection of an AAV2 vector. **(B)** Representative images of GFAP immunostaining in retinal sections from AAV2-CAG-hEDN1 rat retinas 4 wk after intravitreal injection of the AAV2 vector. **(C)** Transcriptional level of *Ednrb* in AAV2-CAG-hEDN1 rat retinas 4 wk after intravitreal injection of an AAV2 vector. GAPDH was used as an internal control. **(D)** Representative images of ET-B immunostaining in retinal sections from AAV2-CAG-hEDN1 rat retinas 4 wk after intravitreal injection of the AAV2 vector. Counterstaining was performed with DAPI (blue). Data are expressed as mean ± SD (n = 6). Scale bar: 50 $\mu$m.

expression after AAV2-CAG-hEDN1 injection. With human *EDN1* overexpression induced by an AAV2 vector on the rat retinal surface, the density of GFAP-positive astrocytes was higher than in controls (Fig 4A). To further investigate the GFAP expression pattern, we performed GFAP immunostaining using retinal sections. 4 wk after AAV2-CAG-hEDN1 injection, GFAP was highly expressed throughout the IPL and INL from the GCL, but was not expressed in retinas injected with a control vector (Fig 4B). Researchers have demonstrated that in a model of endothelin-1-induced chronic optic neuropathy, the endothelin-B receptor (ET-B) is up-regulated in optic nerve astrocytes (Wang et al, 2009). Thus, we tested whether ET-B expression was changed in AAV2-CAG-hEDN1-injected retinas. To evaluate ET-B mRNA expression levels, we used real-time qRT–PCR analysis. In the 4-wk AAV2-CAG-hEDN1 intravitreal injection group, the level of *Ednrb* was 3.4-fold compared with the control group (Fig 4C). Immunostaining images also showed that ET-B expression was clearly increased in the AAV2-CAG-hEDN1-injected retinas, especially in the GCL (Fig 4D). These findings suggest that continuous stimulation of ET-1 promotes ET-B expression in the retinal astrocytes of rats.

### Ischemia-related gene changing in AAV2-CAG-hEDN1 rat retinas

We performed RT–PCR to determine whether the expression of related genes was changed. In our animal model, the expression of not only *Edrb*, but also *Edra* was elevated (2.1-fold) in the 4-wk AAV2-CAG-hEDN1 intravitreal injection group compared with the

control group (Fig 5A). In addition, the transcriptional level of *Aqp4* was also significantly increased (1.8-folds) in the AAV2-CAG-hEDN1 injection group, but *Kcnj10* was not significantly different in the AAV2-CAG-hEDN1 injection group and the control group (Fig 5B and C). Moreover, transcriptional levels of *Ripk1*, *Ripk3*, and *Tnf* were significantly elevated in the 4-wk AAV2-CAG-hEDN1 intravitreal injection group compared with the control group, but this was not significant 2 wk after AAV2 injection, even human *EDN1* was expressed (Figs 5D–F and S3A–I).

### Glutathione reduction in the aqueous humor in AAV2-CAG-hEDN1 rat retinas

Our previous study showed that glaucoma patients have low glutathione levels in the aqueous humor (Sato et al, 2023). Thus, in a final experiment, we measured the glutathione content in the aqueous humor to investigate whether ocular BF dysfunction promotes glutathione reduction. Our high-sensitivity glutathione measurement assay revealed that the glutathione content was lower (by ~45%) in the 4-wk AAV2-CAG-hEDN1 intravitreal injection group compared with the control group (Fig 6). In our BF dysfunction model, intraocular pressure (IOP) did not significantly change with human *EDN1* overexpression (Fig S4). These findings suggest that reduction in the glutathione level in the aqueous humor was caused by ocular BF dysfunction in an IOP-independent manner.

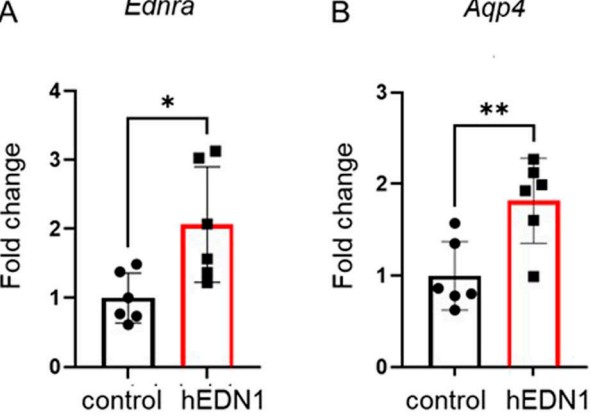

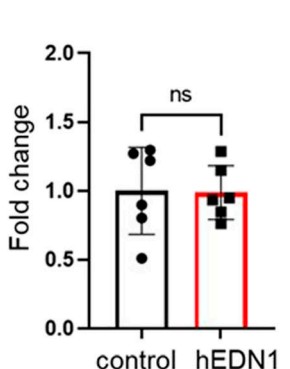

**Figure 5. Gene expression changes in AAV2-CAG-hEDN1 rat retinas 4 wk after intravitreal injection.**
**(A, B, C, D, E, F)** Transcriptional level of *Ednra* (A), *aqp4* (B), *kcnj10* (C), *ripk1* (D), *ripk3* (E), and *Tnf* (F) in AAV2-CAG-hEDN1 rat retinas 4 wk after intravitreal injection of an AAV2 vector. Data are expressed as mean ± SD (n = 6). ns; not significant, *P < 0.05, **P < 0.01. ***P < 0.001.

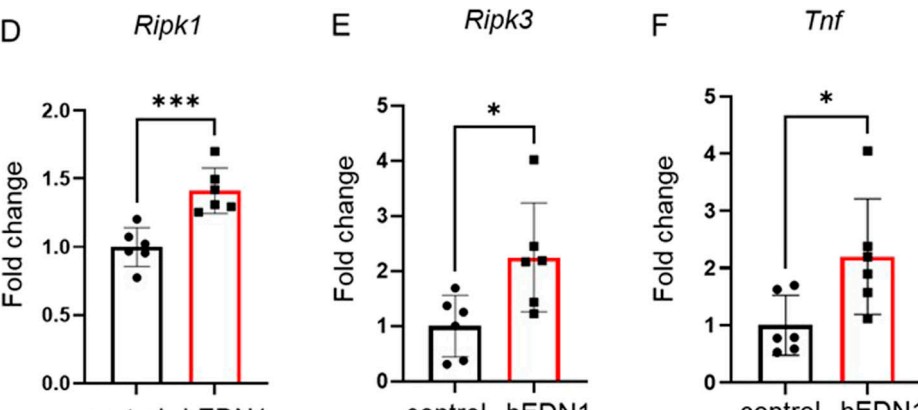

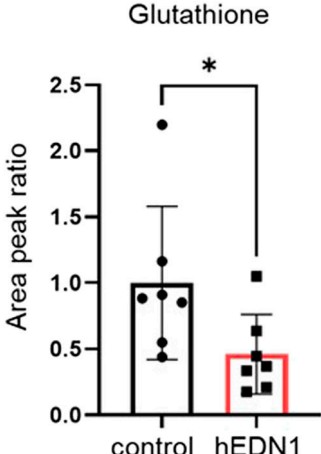

**Figure 6. Glutathione content of the aqueous humor in rats that received an injection of AAV2-CAG-hEDN1.**
Glutathione content was measured with UHPLC-MS/MS in the aqueous humor of rats 4 wk after the intravitreal injection of AAV2-CAG-hEDN1. Area peak values are shown when the amount of GSH in the control is set to one. Data are expressed as mean ± SD (n = 7). *P < 0.05.

# Discussion

Previous studies have shown that the concentration of ET-1 is relevant to the pathogenesis of glaucoma and that high concentrations can lead to the degeneration of RGCs in animal models. The intravitreal injection of ET-1 in rat models results in RGC loss, which affects both anterograde and retrograde axonal transport in RGCs (Stokely et al, 2002; Taniguchi et al, 2006). Rats are commonly used as experimental animal models of the mammalian visual system because they share similar anatomical and developmental patterns with humans, and changes in BF in the normal rat retina can be detected (Remé et al, 1983; Bouhenni et al, 2012). Recently, research evidence is accumulating on the relationship between retinal BF, measured with LSFG, and retinal diseases; however, research using LSFG and disease models has not been sufficient. The point we wish to highlight in this study is that we evaluated retinal BF in an animal model with LSFG, which has recently been used clinically with glaucoma patients. To investigate retinal BF patterns that are similar to those in real glaucoma patients, we attempted to establish a new animal model of chronic, moderate retinal BF reduction after AAV2-CAG-hEDN1 injection. The rat model we developed showed sustained expression of ET-1 using a virus

vector that led to significantly decreased BF and reduced RGC survival in the peripheral retina, as well as astrocyte activation.

A previous study (Taniguchi et al, 2006) revealed that intravitreal injection of 5 pmol of ET-1 into animal eyes, a commonly used method in animal experiments, induced acute, transient retinal vasoconstriction; this was determined by checking fundus photographs. Consistent with this, we confirmed that intravitreal injection of 5 pmol of ET-1 immediately and dramatically reduced retinal BF (~13%) as measured by LSFG; BF almost completely recovered 1 wk after the injection. This transient BF reduction may have been caused by vitreous circulation of the injected ET-1. On the other hand, we found that AAV2-CAG-hEDN1 injection induced moderate, sustained retinal BF reduction, resulting in RGC loss. These results demonstrate that chronic and sustained retinal BF dysfunction also induces RGC death, similarly to models of rapid and transient retinal BF reduction. To investigate the mechanism of RGC death in this disease model, we examined astrocyte activation, which has been mentioned in the past. Astrocytes were originally thought to be inert structural cells in the central nervous system, but they have now emerged as important players in neurodegeneration (Rossi & Volterra, 2009). In glaucoma, abnormal activation and/or proliferation of astrocytes, a process termed astrogliosis, has been observed (Hernandez et al, 2008). At the site of an injury, astrocytes undergo hypertrophy and proliferate, forming a glial scar that limits further damage but also acts as a barrier to regeneration. Reactive gliosis is also characterized by increased glutamate efflux from the astrocytes, which could potentiate excitotoxic cell death of neurons (Sasaki et al, 1997; Maragakis & Rothstein, 2006). This is accomplished through the up-regulation of intermediate filament proteins and GFAP. ET-B, which has been shown to be expressed by astrocytes, can regulate astrocyte hypertrophy in normal and injured optic nerves (Rogers et al, 2003). Increasing ET-B in astrocytes may be a reaction to optic neuropathy (Wang et al, 2009; Murphy et al, 2011; Tonari et al, 2012). In particular, in the optic nerve head, astrocytes proliferate with the stimulation of ET-1, and this is suppressed when there is a lack of functional ET-B (Murphy et al, 2010, 2011). This ET-1-induced astrocyte hypertrophy is dependent on the JNK/c-Jun signaling pathway (Gadea et al, 2008).

Not only astrocytes, but also the retinal vessels are associated with ET-1-induced RGC death. Marola et al found that the specific deletion of endothelin receptor A in retinal vascular cells prevented RGC loss in an ET-1 injury model (Marola et al, 2022). Our AAV2-hEDN1 model elevated *Ednra* expression, suggesting that moderate and chronic vasoconstriction is also important in the development of neurovascular dysfunction in the pathophysiology of glaucoma. In addition, ET-1 directly induces RGC apoptosis, and its neurodegenerative mechanism is involved via JNK-JUN signaling in vitro and in vivo (He et al, 2015; Marola et al, 2020). Moreover, ET-1 increased intracellular NO levels and superoxide in cultured retinal neurons, and ET-1-associated neuronal injury is suppressed by nitric oxide synthase, suggesting that ET-1 can induce NOS activation and oxidative stress in the retina (Oku et al, 2008). In addition, ET-1 is produced by the non-pigmented ciliary epithelium and secreted into the aqueous humor (Lepple-Wienhues et al, 1992), where it can cause contraction of the trabecular meshwork, decreasing the intertrabecular space and thereby increasing outflow resistance and, in turn, IOP (Choritz et al, 2005). However, our

AAV2-hEDN1 model did not promote IOP elevation, demonstrating that ET-1 expression in the retina does not lead to anterior chamber abnormalities, thus indicating that our BF dysfunction model resembles normal-tension glaucoma.

A past study reported that intravitreal ET-2 injection promoted aquaporin4 (AQP4) and Kir4.1 expression in the mouse retina, especially the Müller cells (Alrashdi et al, 2018), while another study reported that AQP4 expression was enhanced in optic nerve astrocytes after IOP elevation in rats (Dibas et al, 2008). AQP4 is a water channel and maintains homeostatic balance in response to osmotic gradients. The alteration of the Aqp4 level in our AAV2-hEDN1 model may reflect a water/osmotic imbalance in the retina, resulting in astrocytic hypertrophy. On the other hand, the *Kcnj10* level, encoding Kir4.1 protein, did not change in our AAV2-hEDN1 model, unlike after ET-2 injection. This difference may be because of the different types of BF dysfunction (i.e., an acute/dramatic or mild/chronic BF reduction). TNF is a potent inflammatory agent in the pathology of glaucoma, and TNF-related necroptosis is promoted by the RIPK1/RIPK3 axis (Nakazawa et al, 2006; Kim et al, 2024). Our data showed that moderate and chronic retinal BF reduction is a trigger that promotes expression of the TNF and RIPK families. A recent study demonstrated that necroptotic cell death increased in retinal ischemia (Tsai et al, 2024). Also, RIPK1/RIPK3 inhibition or RIPK1/3 gene manipulation can prevent RGC loss in several models of glaucoma, such as ischemia/reperfusion, optic nerve crush, and glutamate excitotoxicity (Do et al, 2017; Liu et al, 2022; Kim et al, 2024; Tu et al, 2025). These past findings indicate that glaucomatous BF dysfunction may promote RGC necroptosis via TNF/RIPK1/RIPK3. Interestingly, there are no reports about the relationship between RIPK1/3 and RGC loss in ET-1-induced retinal injury. We emphasize this point as a new finding in the pathology of glaucoma in this study, which we obtained using our established disease model. In addition, we found that glutathione levels in the aqueous humor decreased in our ocular BF model. Interestingly, this rate of reduction is similar to that in a study of glaucoma patients (~45% in the current model and ~47% in the glaucoma study) (Sato et al, 2023). This suggests that BF dysfunction may be a major reason for glutathione reduction and may promote oxidative stress in the eyes of glaucoma patients. In cultures, oxygen glucose deprivation has been found to promote RIP3-dependent necroptosis and lipid peroxidation in a retinal cell line (Ding et al, 2015). Overall, BF dysfunction triggers harmful events such as oxidative stress, inflammation, astrogliosis, and RGC loss via necroptotic mechanisms.

Previous research has also shown that sustained administration of ET-1 with osmotic minipumps could reduce optic nerve BF, resulting in impaired retrograde axonal transport and axonal injury (Chauhan et al, 2004). Chauhan et al measured optic nerve head BF with laser Doppler flowmetry and found that the rate of BF reduction was 68% in their model. Our current AAV2-driven endothelin expression model was consistent with this and showed a similar rate of reduction rate for retinal BF (measured with LSFG), and it also confirmed RGC loss. Cioffi et al administered ET-1 to the retrobulbar space via an osmotic pump in nonhuman primates and found that there was significant axonal loss (Cioffi et al, 2004). That study indicated that chronic ocular BF dysfunction is a disease mechanism in both rodents and primates, suggesting that studies

of BF disorders in rodents can produce findings that are applicable to the study of glaucoma pathology caused by chronic BF disorders. The recent published work by Nordahl et al on an AAV-driven ET-1 overexpression model that was similar to the one used in our study found retinal damage using ERG (Nordahl et al, 2023). The authors demonstrated that their AAV-driven ET-1 expression model started to reduce a-waves and b-waves 8 d after AAV-mEDN1 injection. This indicates that AAV-derived ET-1 had already caused retinal damage 8 d after injection. In contrast, our AAV-hEDN1 model did not result in the loss of RGCs 2 wk after AAV2 injection, although there was a significant reduction in RGCs after 4 wk. This difference may be because of the virus titer; Nordahl et al injected AAV-mEDN1 at $3.2 \times 10^{10}$ gc/eye; however, we injected a smaller virus titer (AAV2-hEDN1 at $5 \times 10^9$ gc/eye). Another possible reason for this difference is that Nordahl et al used albino rats, whereas we used pigmented rats. Moreover, Nordahl et al did not show data on RGC survival and function; thus, the current study was the first to show RGC loss in an AAV2-derived ET-1 expression model that mimics the mild retinal BF reduction in glaucoma patients.

LSFG is a non-invasive method for measuring BF, and it can detect early signs of normal-pressure glaucoma (Sugiyama et al, 2010). In recent years, the involvement of BF in the pathogenesis of glaucoma has been the focus of much research using optical coherence tomography angiography (OCTA), and LSFG is a useful complement to OCTA in that it can measure actual retinal BF in a short time, whereas OCTA can depict vascular structures. In a study using LSFG, it was reported that there was a subgroup of glaucoma patients with reduced BF that preceded optic neuropathy. This suggests that BF insufficiency is a direct modifier of glaucomatous pathology, and it indicates that LSFG is a useful instrument for detecting this risk. This study is the first report to evaluate the association between optic neuropathy and BF insufficiency with LSFG and is of academic value in this regard.

In conclusion, we developed a model of chronic BF dysfunction that resembles the condition of glaucoma patients; this model relies on LSFG for measurement. We demonstrated that chronic retinal BF reduction activates astrocytes, leading to RGC loss, similar to previous ET-1 intravitreal injection models. The current study suggests that the pathology of RGC loss in normal-tension glaucoma may be because of the dysfunction of BF in the retina. Our improved model offers a more accurate representation of the disease process, providing researchers with a valuable tool for studying glaucoma and gaining insights into its mechanisms. Finally, retinal BF is a new therapeutic target to attenuate RGC damage in glaucoma patients.

# Materials and Methods

## Animals

8–10 wk-old male brown Norway rats were purchased from SLC. The animals were maintained at the Tohoku University Graduate School of Medicine under a cycle of 12 h of light and 12 h of dark. The animal experiments in this study adhered to the Association for Research in Vision and Ophthalmology (ARVO) statement on the use of animals in ophthalmic and vision research and were approved by the institutional animal care and use committee of the Guidelines for Animals in Research (approval #2022-056).

## Intravitreal injection

As in our previous research (Takahashi et al, 2023), the rats were anaesthetized with a mixture of ketamine (100 mg/kg body weight) and xylazine (10 mg/kg body weight). The right eye was dilated topically with one drop of 1% tropicamide. Under an operating microscope, a scleral puncture was made ~0.5 mm from the limbus; a 33G needle was then inserted into the puncture, and 5 $\mu$l of $1 \times 10^9$ gc/$\mu$l of AAV2-CAG>hEDN1:WPRE (#VB220331-1386ehf; vector builder) was injected slowly into the vitreous cavity to express human endothelin1. Antibacterial drops and ophthalmic gel were applied after the injection. As a control vector, we injected an AAV2-CAG>ORF_Stuffer:WPRE (#VB220331-1383bvn; vector builder) vector with amino acid 2–83 of the *E. coli* beta-galactosidase sequence inserted.

## Measurement of retinal BF in rats

To quantitatively analyze retinal BF in the rats, we measured the mean blur rate (MBR). This method has been described in detail in previous research (Wada et al, 2016; Takahashi et al, 2023). Briefly, the rats were placed on a horizontal table in a natural position and color maps were produced using the Laser Speckle Flowgraphy-Micro (LSFG-Micro, Softcare Co., Ltd.) device. MBR images were continuously acquired at a frame rate of 30 frames per second for a duration of 4 s, covering a field of view measuring 3.43 × 3.43 mm with a working distance of 110 mm. The accompanying LSFG software (LSFG analyzer, version 3.1.14.0; Softcare Co., Ltd.) then automatically determined vessel-area MBR (MV), and MV was calculated as the average MBR in the vessel area. The measurements were performed three times consecutively without adjusting the position of the rats, and the average MBR was calculated accordingly.

## Immunohistochemistry

### Retinal cryosection immunostaining

Retinal cryosections were prepared as previously described (Sato et al, 2013). Retinal cryosections were washed at 0.05% Tween 20 in PBS (Tw-PBS), then incubated overnight at 4°C with primary antibodies against RBPMS, GFAP, clone GA5, and against endothelin-B receptors. The cryosections were washed in PBS-T and incubated in secondary primers for 1 h at 4°C with Alexa Fluor 488. Counterstaining was performed with DAPI H-1200 (vector). Immunofluorescence images were captured with a fluorescence microscope (BZ-X810; Keyence). Full details of the primary antibodies and secondary antibodies used are reported in Table 1.

### Retinal flat-mount immunostaining

To estimate the RGC and astrocytes density in the whole retina, immunohistochemistry was performed on retinal flat mounts 2 and 4 wk after intravitreal injection. 4 wk after intravitreal injection, the rats were euthanized with an overdose of sevoflurane. Retinal flat mounts were prepared as previously described (Smith & Chauhan,

**Table 1. Antibody list for immunohistochemistry.**

| Antibody | Species | Dilution | Company(Gatalog#) |
|----------|---------|----------|-------------------|
| RBPMS | Rabbit | 1:200 | Abcam (Ab194213) |
| GFAP | Mouse | 1:200 | Merk Millipore (MAB360) |
| ET-B | Rabbit | 1:100 | Alomone labs (AER-002) |
| Alexa 488 | Rabbit | 1:500 | Thermo Fisher Scientific (A11034) |
| Alexa 488 | Mouse | 1:500 | Thermo Fisher Scientific (A21202) |
| Alexa594 | Rabbit | 1:500 | Thermo Fisher Scientific (A21207) |

**Table 2. Assay primer list for qRT–PCR.**

| Gene symbol | Primer ID |
|-------------|-----------|
| Rat Gapdh | Rn01462662_g1 |
| Rat Edn1 | Rn00561129_m1 |
| Human EDN1 | Hs00174961_m1 |
| Rat Kcnj10 | Rn06167081_s1 |
| Rat Aqp4 | Rn01401327_s1 |
| Rat Ednra | Rn00561137_m1 |
| Rat Ednrb | Rn00569139_m1 |
| Rat Ripk1 | Rn01757369_m1 |
| Rat Ripk3 | Rn01481949_g1 |
| Rat Tnf | Rn99999017_m1 |

2018). The retinas were incubated for 4 d at 4°C in primary antibodies against RBPMS and GFAP to identify RGCs and astrocytes, respectively. The retinas were washed in PBS and incubated in secondary primers overnight at 4°C with anti-rabbit Alexa 488 (1:250, 10% donkey serum) and anti-mouse Alexa 488 (1:250, 10% donkey serum). The retinas were then washed in Tw-PBS for 1 h and incubated in PBS for 5 min, mounted with DAPI H-1200 (vector), and cover-slipped. Immunofluorescence images were captured with a fluorescence microscope (BZ-X800; Keyence). All antibodies used in the experiments were diluted with 10% donkey serum. Full details of the primary antibodies and secondary antibodies used are reported in Table 1.

### Quantitative RT–PCR testing of rat retinas

Total RNA was isolated from brown Norway rat retinas by using the FastGene RNA Premium Kit (FG-81050; Nippon Genetic Co. Ltd.) according to the manufacturer's instructions. The reverse transcription was performed by using reverTra Ace qRT–PCR RT Master mix (Code NO. fsq-301S; Toyobo) following the manufacturer's instructions. Real-time PCR was performed with a 7,500 Fast Real-Time PCR system following the manufacturer's instructions. GAPDH primer was used as the internal control. The real-time PCR cycling conditions were as follows: 50°C for 2 min and 95°C for 15 minutes followed by 40 cycles of 94°C for 1 minute and 60°C for 1 min. The comparative CT method (2-ΔΔCT) was used to determine the target gene expression relative to that of GAPDH. Predesigned primers and probes were purchased from Thermo Fisher Scientific Inc. and used in this experiment (listed in Table 2).

### GSH analysis by UHPLC-MS/MS

The rat aqueous humor (2.5 $\mu$l) was transferred to a 1.5 ml sample tube and mixed with normal saline (7.5 $\mu$l). Then, an extract solution (water/methanol/chloroform, 2/5/2, vol/vol/vol, 52 $\mu$l) was added to the tube and mixed with a mixer for 30 s. After centrifuging at 16,000$g$ for 20 min at 4°C, the supernatant (45 $\mu$l) was transferred to another tube. Then, water (40 $\mu$l) was added and the sample was mixed for 30 s and centrifuged at 16,000$g$ for 20 min at 4°C. The supernatant (50 $\mu$l) was transferred to another tube. The sample was dried under vacuum in a centrifuge for 50 min at room temperature. Then, the sample was reconstructed with water (40 $\mu$l) and an N-(9-acridinyl) maleimide (NAM) solution (60 $\mu$l, 0.8 mmol/l, water/acetonitrile, 40/60, vol/vol%) was added and mixed for 30 s. Then, the sample was incubated for 15 min at 23°C for derivatization. Then, the sample (4 $\mu$l) was subjected to ultra-high performance liquid chromatography triple quadrupole mass spectrometry (UHPLC-MS/MS). The UHPLC/MS/MS analysis was performed on an Acquity Ultra Performance LC I-class system (Waters Corp.) interfaced to a Waters Xevo TQ-XS MS/MS system equipped with electrospray ionization. MS/MS was performed using the multiple reaction monitoring mode. The transitions of the precursor ions and product ions, the cone voltage, and the collision energy for the detection of GSH were $m/z$ 582.4, $m/z$ 453.2, 36 V, and 22 eV, respectively. The other settings were as follows: 3.5 kV in the positive ion mode; capillary voltage: 30 V; cone voltage: 50 V; source offset; source temperature: 150°C; 150 l/hr cone gas (N2) flow rate; desolvation temperature: 450°C; 1,000 l/hr desolvation gas flow; 0.15 min/ml collision gas flow; and 7.0 bar nebulization gas (N2) flow. LC separation was performed using a reverse-phase column (Acquity UPLC BEH C18 1.7 $\mu$m, 2.1 × 50 mm with C18 Vanguard Pre-Column) with a gradient elution using solvent A (10 mmol/l ammonium bicarbonate) and B (acetonitrile) at 0.45 ml/min: 0.1–2.0% B from 0.0 to 2.8 min, 2.0–30% B from 2.0 to 5.0 min, 30–99% B from 5.0 to 6.0 min, 99% B for 2.0 min, 99–0.1% B from 8.0 to 8.1 min, and 0.1% B for 2.0 min. The oven temperature was 45°C. The data were collected using MassLynx v4.2 software (Waters Corp.), and the peak area of GSH was calculated by Traverse MS (Reifycs Inc.) for further statistical analysis.

### Evaluation of IOP

Rat IOP was measured with a rebound tonometer (icare) under anaesthesia with isoflurane, as previously described (Ikuta et al, 2017). IOP was measured before AAV injection and 4 wk after AAV injection.

### Statistical analysis

The data are expressed as the mean ± SD. Statistical significance was calculated with GraphPad Prism software (version 9) and the unpaired $t$ test for comparisons of pairs of groups. We considered $P < 0.05$ to be significant.

## Supplementary Information

# Acknowledgements

We thank Mr. Tim Hilts for editing this document, and Ms. Junko Sato, Ms. Rieko Kamii, and Ms. Mayumi Suda for technical assistance. This work was partly supported in part by JST Grant Number JPMJPF2201 (T Nakazawa), JSPS KAKENHI Grant Number 23K27316 (D Saigusa) and ROHTO Pharmaceutical Co., Ltd.

## Author Contributions

G Shi: investigation.
K Sato: conceptualization, formal analysis, supervision, investigation, and writing—original draft, review, and editing.
N Takahashi: investigation.
M Ohno-Ohishi: investigation.
N Murayama: investigation.
C Yamaguchi: investigation and writing—original draft.
D Saigusa: data curation, funding acquisition, investigation, and writing—review and editing.
T Nakazawa: conceptualization, supervision, funding acquisition, and writing—review and editing.

## Conflict of Interest Statement

The authors declare that they have no conflict of interest.

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
