## [Reviewer comments · Life Science Alliance]

Life Science Alliance

AAV2-driven endothelin induces chronic reduced retinal blood flow/retinal ganglion cell loss in rats

Ge Shi, Kota Sato, Nana Takahashi, Michiko Ohno-Ohishi, Namie Murayama, Chiaki Yamaguchi, Daisuke Saigusa, and Toru Nakazawa

DOI: <https://doi.org/10.26508/lsa.202403087>

Corresponding author(s): Toru Nakazawa, Tohoku University

Review Timeline:	Submission Date:	2024-10-11
	Editorial Decision:	2024-11-22
	Revision Received:	2025-04-10
	Editorial Decision:	2025-04-15
	Revision Received:	2025-04-30
	Accepted:	2025-05-01

Scientific Editor: Tim Fessenden

Transaction Report:

November 22, 2024

Re: Life Science Alliance manuscript #LSA-2024-03087

Prof. Toru Nakazawa
Tohoku University
Department of Ophthalmology, Tohoku University Graduate School of Medicine
1-1, Seiryō-cho
Sendai 980-8574
Japan

Dear Dr. Nakazawa,

Thank you for submitting your manuscript entitled "New disease model of chronic retinal blood flow dysfunction using AAV2-driven endothelin overexpression to promote RGC loss in rats" to Life Science Alliance. The manuscript was assessed by expert reviewers, whose comments are appended to this letter. We invite you to submit a revised manuscript addressing the Reviewer comments.

Thank you for this interesting contribution to Life Science Alliance. We are looking forward to receiving your revised manuscript.

Sincerely,

B. MANUSCRIPT ORGANIZATION AND FORMATTING:

Reviewer #1 (Comments to the Authors (Required)):

The manuscript by Shi et al. demonstrates ET-1 mediated decrease in blood flow in a rat model of AAV-2 mediated expression of ET-1 in the retina. The manuscript is well written providing good experimental details and compelling data to demonstrate endothelin-mediated sustained blood flow reduction and RGC loss occurring following a chronic AAV-2 mediated upregulation of ET-1 expression. There are few points that merit attention:

1. In the introduction section, on line 72, the wording should be changed from "...retinal BF is considered to be a cause of this" to "retinal BF is considered to be a cause of these neurodegenerative changes."
2. The statement on line 84 and 85 about LSFG measured retinal blood flow decreased in normal tension glaucoma and open angle glaucoma needs to be referenced.
3. The authors used only male Brown Norway rats. It would be important to do the study in female rats to see if the similar changes in blood flow and RGC viability are observed.
4. In line 142, the expanded form of Tw-PBS must be provided.
5. In line 145, the word "primers" must be changed to "antibody". Similarly, on line 159, the word "primers" must be changed to "antibody".
6. In the results section, on line 206, it is unclear what AAV-2-CAG-stuffer refers to. This must be clarified.
7. In line 213, the units of human EDN1 concentration must be provided. Is this pg/ml or pM? Similarly, the concentration of rat EDN1 must be stated on line 217.
8. On line 242, a better description should be provided about ion and water disturbances
9. In Figure 4, the images provided do not suggest a 10-fold increase in GFAP protein levels AAV2-CAG-hEDN1 injected rats. A more representative image must be provided.
10. In line 315, the authors must quote the work of Sasaki et al. (1997) Endothelin evokes efflux of glutamate in cultures of rat astrocytes. *J Neurochem.* 68(5):2194-2200.

Reviewer #2 (Comments to the Authors (Required)):

The authors do not have a new model for glaucoma and retinal ischemia: this was reported by Nordahl et al, 2023. They both use AAV2/2 for ET1-delivery. The use of AAV-driving ET-1 is a revised model of the Cioffi model from 2007, and Yorio (2006)-direct injection of ET1 retrobulbar and/IVT. They even use the same Woodchuck promotor (WPRE) as the Nordahl paper.

What is new is the use of laser speckle imaging, and actual RGC counts. They also used somewhat different dosing. This is useful information. I do not have a problem with their work, their techniques, or their results. But the title should be changed to something that reflects the fact that this work is an extension of Nordahl et al (their reference #42), and this should also be introduced in the introduction, rather than simply in the discussion. They can then discuss the changes and differences that they see and how they correlate with the results from Nordahl.

Manuscript #: LSA-2024-03087

Title: AAV2-driven endothelin induces chronic reduced retinal blood flow/retinal ganglion cell loss in rats

Authors: Ge Shi et al.

Author's general reply: We appreciate the valuable comments from the reviewers. In response to the reviewers' comments, we have made changes to the manuscript, which are highlighted in yellow, and modified the figures. We would also like to change the title of this manuscript from "New disease model of chronic retinal blood flow dysfunction using AAV2-driven endothelin overexpression to promote RGC loss in rats" to "AAV2-driven endothelin induces chronic reduced retinal blood flow/retinal ganglion cell loss in rats."

We added more time points to the reported LFSG parameters to add clarity to their presentation, shown in a new figure (figure 2B) and in the text on page 12, lines 254-256.

We also made revisions to increase the sample size for RBPMS-positive RGC count and changed the y-axis caption from "RBPMS-positive RGC number" to "RGC survival rate as 100% of control," because we combined two individual datasets with normalization (new Figure 3 and new Supplementary Figure 2B).

We also added several figures to emphasize and clarify the disease mechanism in our AAV2-hEDN1 model. Specifically, we added additional data showing an increase of necroptosis markers in the retina and a decrease in glutathione in the aqueous humor due to chronic ocular blood flow reduction (new Figures 5 and 6). In addition, we added data on intraocular pressure (IOP) to confirm that our AAV2-hEDN1 model modulates IOP (new Supplementary Figure 4).

Replies to reviewer 1

Major points:

1. In the introduction section, on line 72, the wording should be changed from "...retinal BF is considered to be a cause of this" to "retinal BF is considered to be a cause of these neurodegenerative changes."

ANSWER:

We thank the reviewer for the valuable comments. We have revised this sentence as suggested. We also revised "retinal BF" to "ocular BF," because LSFG also measures blood flow in the optic nerve and choroid, not only the retina.

2. The statement on line 84 and 85 about LSFG measured retinal blood flow decreased in normal tension glaucoma and open angle glaucoma needs to be referenced.

ANSWER:

As suggested by the reviewer, we have added appropriate citations for this statement, as follows: (Kiyota et al., 2018)(Gu et al., 2021)(Kiyota et al., 2021)(Yamaguchi et al., 2024).

3. The authors used only male Brown Norway rats. It would be important to do the study in female rats to see if the similar changes in blood flow and RGC viability are observed.

ANSWER:

We thank the reviewer for this meaningful comment. We used male rats for the following reasons: 1) Our review of previously published studies revealed that the majority of studies used male rats or mice. 2) We have used male rats in LSFG experiments in previous reports, and we wanted to ensure comparability with previous studies by using established methods (Takahashi et al., 2023). 3) We consider that female rats are more susceptible to the effects of sex hormones due to the sexual cycle. Using male rats eliminates sex hormone–

dependent effects as much as possible. However, as the reviewer points out, it is also important to compare female and male animals. This point will need to be clarified in future research.

4. In line 142, the expanded form of Tw-PBS must be provided.

ANSWER:

We thank the reviewer for the recommendations. As suggested, we have expanded this term to "0.05% Tween 20 in PBS (Tw-PBS)."

5. In line 145, the word "primers" must be changed to "antibody". Similarly, on line 159, the word "primers" must be changed to "antibody".

ANSWER:

We appreciate the reviewer's thoughtful comment. We have made these changes.

6. In the results section, on line 206, it is unclear what AAV-2-CAG-stuffer refers to. This must be clarified.

ANSWER:

Thank you for pointing out that our explanation was inadequate. We have added detailed information on this negative control vector to the Methods section, as follows (page 6, line 131-133):

As a control vector, we injected an AAV2-CAG>ORF_Stuffer:WPRE (vector builder #VB220331-1383bvn) vector with amino acid 2-83 of the E. coli beta-galactosidase sequence inserted.

7. In line 213, the units of human EDN1 concentration must be provided. Is this pg/ml or pM? Similarly, the concentration of rat EDN1 must be stated on line 217.

ANSWER:

We appreciate the important question raised by the reviewer. Our results for *human EDN1* in this study are gene expression levels, not ET-1 protein concentrations. We emphasize that AAV induced sufficient expression of exogenous ET-1 with a *human-specific* endothelin qPCR primer in the AAV-injected rats' eyes. Since ET-1 induced with AAV is of human origin, we can measure exogenous *EDN1* (human) and endogenous *Edn1* (rat) by using human-specific primers. Also, we attempted to quantify AAV-induced ET-1 levels with ELISA and found no difference between AAV2-hEDN1-treated and control groups. We speculate that this is probably because the ET-1 induced by AAV2-hEDN1 was specific and *limited to the RGCs*, and no clear difference could be detected when the *entire retina* was used as a sample. In other words, the endothelin induced by AAV had a localized effect in the RGCs. It may therefore have been impossible to detect this elevated endothelin expression in a localized area when the entire retina was sampled.

8. On line 242, a better description should be provided about ion and water disturbances

ANSWER:

We completely agree with this point raised by the reviewer. We have added the following sentences to the Discussion section about the ion channel and water channel as follows (page 19, lines 396-406):

A past study reported that intravitreal ET-2 injection promoted aquaporin4 (AQP4) and Kir4.1 expression in the mouse retina, especially the Müller cells(Alrashdi et al., 2018). Also, AQP4 expression was enhanced in optic nerve astrocytes following IOP elevation in rats(Dibas et al., 2008). AQP4 is a water channel and maintains the homeostatic balance in to response to osmotic gradients. The alteration of the *Aqp4* level in our AAV2-hEDN1 model may reflect a water/osmotic imbalance in the retina, resulting in astrocytic hypertrophy. On the other hand, the *Kcnj10* level, encoding Kir4.1

protein, did not change in our AAV2-hEDN1 model, unlike after ET-2 injection. This difference may be due to the different types of blood flow dysfunction (i.e., an acute/dramatic or mild/chronic blood flow reduction).

9. In Figure 4, the images provided do not suggest a 10-fold increase in GFAP protein levels AAV2-CAG-hEDN1 injected rats. A more representative image must be provided.

ANSWER:

We thank the reviewer for pointing out the discrepancy in our data. A previous, related study evaluated astrocyte activation with GFAP immunostaining using retinal flat mounts, but did not use qPCR. Thus, we performed GFAP staining in retinal flat-mounts and replaced the qPCR data in the revised version of our paper. The immunostaining data showed that GFAP-positive astrocytes were densely and intensely stained 4 weeks after AAV2-hEDN1 injection (new Figure 4A). In addition, we also replaced the immunohistochemistry data with anti-GFAP antibody data from retinal cryosections to provide a more representative image (New Figure 4B).

Thus, we added a new Figure 4 and revised the Results section as follows (page 14, lines 286-288):

With hEDN1 overexpression induced by an AAV2 vector on the rat retinal surface, the density of GFAP-positive astrocytes was higher than in controls (Fig. 4A).

10. In line 315, the authors must quote the work of Sasaki et al. (1997) Endothelin evokes efflux of glutamate in cultures of rat astrocytes. *J Neurochem.* 68(5):2194-2200.

ANSWER:

We appreciate the reviewer's extremely helpful remarks. The paper they have introduced us to presents data that are very significant for explaining the mechanism of RGC failure in our model. We agree that it is important to cite this paper in the Discussion section and have done so in the revised paper (Sasaki et al., 1997)(Page 17, Line 369).

Replies to reviewer 2

The authors do not have a new model for glaucoma and retinal ischemia: this was reported by Nordahl et al, 2023. They both use AAV2/2 for ET1-delivery. The use of AAV-driving ET-1 is a revised model of the Cioffi model from 2007, and Yorio (2006)-direct injection of ET1 retrobulbar and/IVT. They even use the same Woodchuck promotor (WPRE) as the Nordahl paper.

What is new is the use of laser speckle imaging, and actual RGC counts. They also used somewhat different dosing. This is useful information. I do not have a problem with their work, their techniques, or their results. But the title should be changed to something that reflects the fact that this work is an extension of Nordahl et al (their reference #42), and this should also be introduced in the introduction, rather than simply in the discussion. They can then discuss the changes and differences that they see and how they correlate with the results from Nordahl.

ANSWER:

We thank the reviewer for the meaningful comment. This review comment raises an extremely important point about this study, and we need to articulate in what ways this study is novel compared to previous studies. As pointed out by the reviewer, Nordahl et al also used AAV2-hEDN1. Points of difference with our study are as follows: 1) They used albino SD rats, whereas we used pigmented BN rats, because it is hard to use LSFG in SD rats due to fundus reflection. 2) We quantified ocular blood flow with LSFG, a technology that is used clinically in the examination of patients with glaucoma. 3) We measured real RGC damage

by counting RBPMS-positive cells. We have added these points to the Introduction and Discussion sections, as follows (page 5, lines 101-104):

Recently, Nordahl et al reported that AAV2-derived ET-1 can cause continuous mild vascular dysfunction, resulting in retinal degeneration (Nordahl et al., 2023). However, the use of LSFG in animal models of ocular BF disorders and the relationship with actual RGC loss have not been fully investigated.

And as follows (page 20-21, lines 435-454):

Cioffi et al administered ET-1 to the retrobulbar space via an osmotic pump in nonhuman primates and found that there was significant axonal loss (Cioffi et al., 2004). That study indicated that chronic ocular blood flow dysfunction is a disease mechanism in both rodents and primates, suggesting that studies of blood flow disorders in rodents can produce findings that are applicable to the study of glaucoma pathology caused by chronic blood flow disorders.

The recent published work by Nordahl et al on an AAV-driven ET-1 overexpression model that was similar to the one used in this study found retinal damage using ERG (Nordahl et al., 2023). The authors demonstrated that their AAV-driven ET-1 expression model started to reduce a-waves and b-waves 8 days after AAV-mEDN1 injection. This indicates that AAV-derived ET-1 had already caused retinal damage 8 days after injection. In contrast, our AAV-hEDN1 model did not result in the loss of RGCs 2 weeks after AAV2 injection, although there was a significant reduction in RGCs after 4 weeks. This difference may be due to virus titer; Nordahl et al injected AAV-mEDN1 at 3.2×10^{10} gc/eye; however, we injected a smaller virus titer (AAV2-hEDN1 at 5×10^9 gc/eye). Another possible reason for this difference is that Nordahl et al used albino rats, whereas we used pigmented rats. Moreover, Nordahl et al did not show data on RGC survival and function; thus, the

current study is the first to show RGC loss in an AAV2-derived ET-1 expression model that mimics the mild retinal BF reduction in glaucoma patients.

We were not able to find the paper mentioned by the reviewer about Yorio (2006)-direct injection of ET1 retrobulbar and/IVT and therefore could not include it in the Discussion section. We apologize for that.

<Reference>

- Alrashdi SF, Deliyanti D, Talia DM, Wilkinson-Berka JL. 2018. Endothelin-2 Injures the Blood–Retinal Barrier and Macrogial Müller Cells: Interactions with Angiotensin II, Aldosterone, and NADPH Oxidase. *Am J Pathol* [Internet] 188:805–817. Available from: <https://doi.org/10.1016/j.ajpath.2017.11.009>
- Cioffi GA, Wang L, Fortune B, Cull G, Dong J, Bui B, Van Buskirk EM. 2004. Chronic ischemia induces regional axonal damage in experimental primate optic neuropathy. *Arch Ophthalmol* 122:1517–1525.
- Dibas A, Yang M, He S, Bobich J, Yorio T. 2008. Changes in ocular aquaporin-4 (AQP4) expression following retinal injury. 4:1770–1783.
- Gu C, Li A, Yu L. 2021. Diagnostic performance of laser speckle flowgraphy in glaucoma: a systematic review and meta-analysis. *Int Ophthalmol* [Internet] 41:3877–3888. Available from: <https://doi.org/10.1007/s10792-021-01954-3>
- Kiyota N, Kunikata H, Shiga Y, Omodaka K, Nakazawa T. 2018. Ocular microcirculation measurement with laser speckle flowgraphy and optical coherence tomography angiography in glaucoma. *Acta Ophthalmol* 96:e485–e492.
- Kiyota N, Shiga Y, Omodaka K, Pak K, Nakazawa T. 2021. Time-Course Changes in Optic Nerve Head Blood Flow and Retinal Nerve Fiber Layer Thickness in Eyes with Open-angle Glaucoma. *Ophthalmology* [Internet] 128:663–671. Available from: <https://doi.org/10.1016/j.ophtha.2020.10.010>
- Nordahl KML, Fedulov V, Holm A, Haanes KA. 2023. Intraocular Adeno-Associated Virus-Mediated Transgene Endothelin-1 Delivery to the

- Rat Eye Induces Functional Changes Indicative of Retinal Ischemia—A Potential Chronic Glaucoma Model. *Cells* 12.
- Sasaki Y, Takimoto M, Oda K, Früh T, Takai M, Okada T, Hori S. 1997. Endothelin evokes efflux of glutamate in cultures of rat astrocytes. *J Neurochem* 68:2194–2200.
- Takahashi N, Sato K, Kiyota N, Tsuda S, Murayama N. 2023. OPEN A ginger extract improves ocular blood flow in rats with endothelin - induced retinal blood flow dysfunction. *Sci Rep* [Internet]:1–8. Available from: <https://doi.org/10.1038/s41598-023-49598-w>
- Yamaguchi C, Kiyota N, Himori N, Omodaka K, Tsuda S, Nakazawa T. 2024. Differentiating optic neuropathies using laser speckle flowgraphy: Evaluating blood flow patterns in the optic nerve head and peripapillary choroid. *Acta Ophthalmol*:49–57.

April 15, 2025

RE: Life Science Alliance Manuscript #LSA-2024-03087R

Prof. Toru Nakazawa
Tohoku University
Department of Ophthalmology
1-1, Seiryomachi, aoba-ku
Sendai 9808574
Japan

Dear Dr. Nakazawa,

Thank you for submitting your revised manuscript entitled "AAV2-driven endothelin induces chronic reduced retinal blood flow/retinal ganglion cell loss in rats". After carefully examining this revision, the reviewer comments, and your rebuttal we have determined that further reviewer input is not needed. We would be happy to publish your paper in Life Science Alliance pending final revisions necessary to meet our formatting guidelines.

- please be sure that the authorship listing and order is correct.
- please upload all figure files as individual ones and remove them from main manuscript file, including the supplementary figure files; all figure legends should only appear in the main manuscript file after the 'References'.
- please add a Summary Blurb/Alternate Abstract in our system.
- please add ORCID ID for corresponding author--you should have received instructions on how to do so.
- please add the X and Bluesky handles of your host institute/organization as well as your own or/and one of the authors in our system.
- please consult our manuscript preparation guidelines <https://www.life-science-alliance.org/manuscript-prep> and make sure your manuscript sections are in the correct order.
- please move Tables at the end of manuscript file.
- please add a Conflict of Interest statement to your main manuscript text.
- please add an Author Contributions section to your main manuscript text.
- please include labels C-D in the caption of Figure 3 and Supplementary Figure 2, as they are present in the figure itself.
- please add a callouts for Supplementary Figure 2A-D and Supplementary Figure 3A-I to your main manuscript text.
- please include scale bars for Figures 1A, 2A, 3A, 4A,B,D, S1A-B, and S2A.

A. FINAL FILES:

-- Summary blurb (enter in submission system): A short text summarizing in a single sentence the study (max. 200 characters including spaces). This text is used in conjunction with the titles of papers, hence should be informative and complementary to

the title. It should describe the context and significance of the findings for a general readership; it should be written in the present tense and refer to the work in the third person. Author names should not be mentioned.

B. MANUSCRIPT ORGANIZATION AND FORMATTING:

Sincerely,

May 1, 2025

RE: Life Science Alliance Manuscript #LSA-2024-03087RR

Prof. Toru Nakazawa
Tohoku University
Department of Ophthalmology, Tohoku University Graduate School of Medicine
1-1, Seiryō-cho
Sendai 980-8574
Japan

Dear Dr. Nakazawa,

Thank you for submitting your Research Article entitled "AAV2-driven endothelin induces chronic reduced retinal blood flow/retinal ganglion cell loss in rats". It is a pleasure to let you know that your manuscript is now accepted for publication in Life Science Alliance. Congratulations on this interesting work.

DISTRIBUTION OF MATERIALS:

Again, congratulations on a very nice paper. I hope you found the review process to be constructive and are pleased with how the manuscript was handled editorially. We look forward to future exciting submissions from your lab.

Sincerely,
